# Diet modulates strongyle infection and microbiota in the large intestine of horses

**Noémie Laroche** [1,2]*, **Pauline Grimm**[1], **Samy Julliand**[1], **Gabriele Sorci**[2]

**1** Lab To Field, 26 bd Docteur Petitjean, Dijon, France, **2** Biogéosciences, CNRS UMR 6282, Université de Bourgogne Franche-Comté, 6 Boulevard Gabriel, Dijon, France

* noemie.laroche@lab-to-field.com

**Data Availability Statement:** All relevant data are within the manuscript and its Supporting information files.

**Funding:** Multifolia, the French Association Nationale Recherche Technologie (ANRT : 2020/

## Abstract

The use of anthelminthic drugs has several drawbacks, including the selection of resistant parasite strains. Alternative avenues to mitigate the negative effects of helminth infection involve dietary interventions that might affect resistance and/or tolerance by improving host immunity, modulating the microbiota, or exerting direct anthelmintic effects. The aim of this study was to assess the impact of diet on strongyle infection in horses, specifically through immune-mediated, microbiota-mediated, or direct anthelmintic effects. Horses that were naturally infected with strongyles were fed either a high-fiber or high-starch diet, supplemented with either polyphenol-rich pellets (dehydrated sainfoin) or control pellets (sunflower and hay). When horses were fed a high-starch diet, they excreted more strongyle eggs. Adding sainfoin in the high-starch diet reduced egg excretion. Additionally, sainfoin decreased larval motility whatever the diet. Moreover, the high-starch diet led to a lower fecal bacterial diversity, structural differences in fecal microbiota, lower fecal pH, lower blood acetate, and lower hematocrit compared to the high-fiber diet. Circulating levels of Th1 and Th2 cytokines, lipopolysaccharides, procalcitonin, and white blood cells proportions did not differ between diets. Overall, this study highlights the role of dietary manipulations as an alternative strategy to mitigate the effect of helminth infection and suggests that, in addition to the direct effects, changes in the intestinal ecosystem are the possible underlying mechanism.

## Introduction

Gastrointestinal helminths are widespread parasites, infecting humans, livestock and wildlife [1]. According to the World Health Organization, soil-transmitted helminths infect 1.5 billion people [2]. These infections can incur substantial morbidity including malnutrition and stunting, particularly when the intensity of infection is high. In livestock, helminth-associated morbidity includes reductions in productivity (e.g., growth, fertility, milk yield, carcass weight) and animal welfare [3, 4]. In horses, two groups of gastrointestinal nematodes are particularly common: large strongyles and cyathostomins. Cyathostomins, with the genera *Cyathostomum*, *Cylicocyclus*, *Cylicostephanus* and *Coronocyclus* being the most abundant, can cause severe symptoms such as weight loss, diarrhea and colitis, which can potentially lead to death [4, 5].

1750) and the Fonds Européen de DEveloppement Régional (FEDER : BFC000794) provided financial support for this study. The funders had no role in study design, data collection and analysis, decision to publish, or preparation of the manuscript.

**Competing interests:** The authors declare no conflict of interest.

Gastrointestinal helminth infections have a significant economic impact in both livestock and horses [6]. In ruminants, recent work has estimated that the combined cost of infection with gastrointestinal nematodes, the trematode *Fasciola hepatica*, and the lungworm *Dictyocaulus viviparus* sums up to 1.8 billion euros in 18 European countries [7]. Due to the public health consequences of infection in humans, the negative impacts on animal health and welfare and the huge economic loss, particularly in the livestock industry, anthelmintic drugs are commonly administered as a deworming treatment. However, the widespread use of anthelmintic drugs, especially in the livestock and equine industries has produced a series of negative outcomes. First, the selection pressure exerted by the drugs has promoted the evolution and spread of resistant strains. Resistance to benzimidazole is very widespread in gastrointestinal helminths infecting ruminants [8] and horses [9]. Resistance to macrocyclic lactone is also very common in ruminants [8] and it has increased in frequency over the last years in horses [10]. Second, anthelmintic residues are released into the environment and can have toxic effects on terrestrial and aquatic species [11]. Third, anthelmintics can disrupt the gut microbiota and compromise the services provided by a healthy microbiota [12, 13]. For example, a single treatment with moxidectin and praziquantel induces a reduction in the diversity of the gastrointestinal microbiota in non-infected horses, although host factors (independently of the treatment) appear also to be important [12]. Finally, the cost of anthelmintic drugs has been estimated to account for approximately 20% of the total economic losses caused by helminth infections in livestock [3]. For these reasons, it is necessary to explore alternative and more sustainable strategies to control and manage helminth infections.

Dietary interventions have been suggested to have the potential to modulate helminth infections through various pathways (Fig 1). Some dietary components might have direct anthelmintic properties. Secondary plant metabolites have been demonstrated to have anthelmintic properties in *in vitro* and *in vivo* studies, although the mechanism of action is not yet fully understood. The anthelmintic properties of polyphenols have been investigated in ruminants [14, 15] with evidence suggesting that polyphenols can reduce the intensity of infection (the number of adult worms) and worm fecundity, which can limit the transmission to other hosts [16].

Diet can also indirectly control helminth infection by modulating host immunity and shaping the diversity and composition of the gut microbiota. The macronutrients and micronutrients in the diet can affect several immune parameters, contributing to an improved host resistance and tolerance to helminths. For example, polyphenols may enhance mucosal integrity and host immunity [17]. Indeed, some studies have shown that helminth-infected pigs and sheep that were fed a diet supplemented with polyphenols had higher numbers of Th2-associated mucosal eosinophils and mast cells [18, 19]. Additionally, studies in livestock have shown that amino acids supplementation increases the activation of T-cells, Natural Killer cells and macrophages, and enhances antibody production [20]. Sheep infected with the gastrointestinal nematodes *Haemonchus contortus* and *Trichostrongylus colubriformis* exhibited a greater peripheral eosinophilia and a higher count of globular leukocytes in the abomasum when feeding a cysteine-supplemented diet [21]. Similarly, there is substantial evidence demonstrating that diet plays a major role in the acquisition and maintenance of a diverse and healthy microbiota [e.g., 22, 23]. Dietary interventions can indirectly affect gastrointestinal helminth infection by altering the diversity and composition of the microbiota. Studies have shown reciprocal (immune-mediated) effects between the microbiota and gut-dwelling helminths [24, 25]. However, due to the inconsistencies in reported effects across model systems (host/parasite species), it is difficult to have a general picture of how helminths shape microbiota diversity and vice-versa. Nevertheless, a recent meta-analysis conducted on human

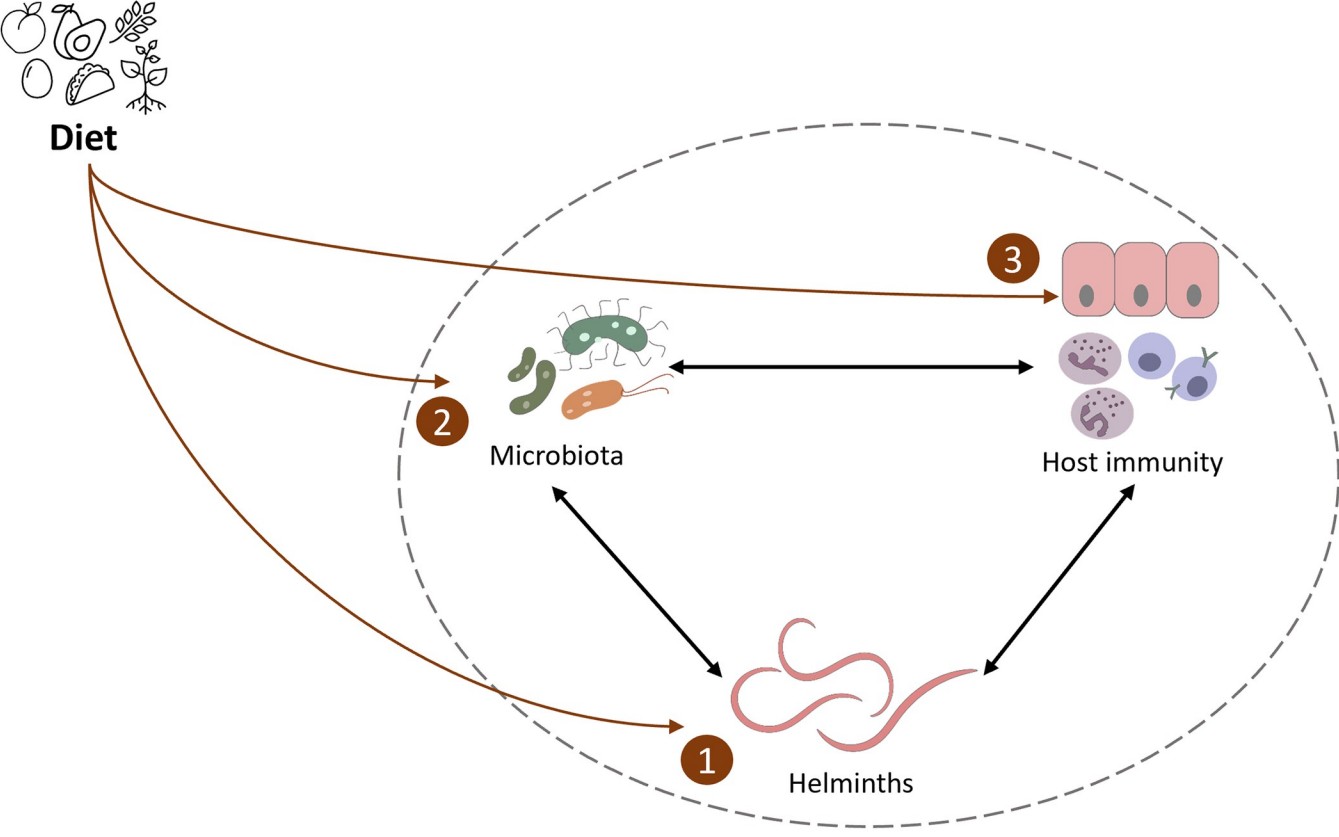

**Fig 1. Diet can have direct (1) and indirect (2 and 3) effects on helminth infection.** Direct effects include food with anthelmintic properties (e.g., secondary plant metabolites) that might reduce the likelihood of establishment or the fecundity and survival of worms within the host. Indirect effects include the modulation of the gut microbiota and immunity.

studies has shown a consistent positive effect of helminth infection on the richness and α-diversity of the gut microbiota [26].

The aim of this work was to investigate the potential of dietary interventions as an alternative to anthelmintic drugs in horses. The study also aimed to explore the possible mechanisms of action, including direct effects and indirect effects through immunity and the microbiota. Previous research has shown that the inclusion a polyphenol-rich plant (sainfoin, *Onobrychis viciifolia*) in the diet of naturally infected horses can reduce the excretion of eggs in the feces and the motility of infective larvae [27]. Additionally, evidence suggests that dietary composition can affect the diversity, composition, and function of the microbiota in the large intestine of horses [28]. In particular, a high-starch diet can lead to dysbiosis and increased intestinal permeability [29, 30].

In this study, two diets (high-fiber and high-starch) were fed to horses, with or without supplementation of a polyphenol-rich plant (sainfoin), and we investigated the effects of diet and supplementation on: i) the amount of strongyle eggs excreted in the feces and the motility of larvae, ii) the diversity, composition, function and activity of the large intestine microbiota, iii) a panel of systemic immune markers, and host health. Specifically, it was predicted that i) adding a polyphenol-rich plant to the diet would result in a reduction in egg excretion and larval motility; ii) feeding on a high starch diet would lead to dysbiosis and an increase in markers of inflammation; iii) adding a polyphenol-rich plant may improve the inflammatory status.

## Materials and methods

### Ethics statement

The study was conducted in the experimental stables of the company Lab To Field (Créancey, France) by members of the company staff. The study was approved by the Committee on the Ethics of Animal Experiments of the University of Burgundy, and authorized by the French Ministry of Research with the permit number #30448–2021031111512359.

### Animals and management

Twelve adult "Trotteurs Français" geldings naturally infected with strongyles at the beginning of the trial were included in this study, which was conducted from May to September 2021. All horses were up to date with their vaccinations, following the protocol established by the in-house veterinarian. Eleven horses were last dewormed 7 months prior to the start of the study and one horse was last dewormed with ivermectin (Alverin, Zoetis, 0.2 mg/kg) one year prior to the start of the study. Horses were individually housed in 13.3 m$^2$ boxes, were exercised for 1 hour per day in an automatic walker at a speed of 6 km/h and spent 4.5 hours per day in overgrazed paddocks. The horses had free access to water and salt lick blocks (Solsel$^®$, K+S Minerals and Agriculture GmbH, Kassel, Germany). Horses were observed daily to make sure that there was no signs of discomfort or suffering associated with the dietary treatments.

### Experimental design and dietary treatments

At the beginning of the trial, horses with similar fecal egg counts (FEC), age (years), body weight (kg) and body condition score [31] (S1 Table) were allocated to four dietary treatments in a Latin square design with four experimental periods. During each experimental period which lasted 21 days, horses were fed either with a high-fiber diet (hay, HF) or a high-starch diet (hay and barley, HS) and received a supplement of either polyphenol rich plant (sainfoin pellets, Perly cultivar, Equifolia$^®$, Multifolia, Viapres Le Petit, France, SF) or a mix of sunflower and hay pellets (as a control, CONT). Sainfoin contained 1.5% DM of condensed tannins and 2.8% DM of total phenols (Folin method N-AOEN/M/119, Inovalys Laboratory, Nantes, France); tannins and phenols were considered negligible in the mix of sunflower meal and hay pellets. At the end of each experimental period, horses were fed with hay *ad libitum* for 21 days (washout periods). The experimental design is described in S1 Fig (S1 Fig). Analysis of the composition of the raw materials was performed by DairyOne (Equine Complete analysis; Ithaca, USA) and the four dietary treatments were formulated to be iso-caloric and iso-nitrogenous (Table 1) and to meet the energy requirements for horses moderately exercised (National Research Council, 2007). A gradual decrease in hay and a gradual increase of barley were carried out on the first 5 days of each experimental period when horses received the HS diet. Food was provided in two equal meals per day, at 08:00 and 16:45.

### Strongyle fecal egg count and larval motility

Fecal samples were collected from each horse before the start of the experimental diets ($D_0$), and then on a weekly basis (from $D_7$ to $D_{21}$) by rectal grab. To assess the number of strongyle eggs excreted in the feces, 42 ml of NaCl-saturated water (300 g/L) were added to three grams of feces and mixed. The suspension was then filtered through a sieve (500 μm pore size) and the filtrate was inserted into the two chambers of a McMaster slide with a detection limit of 50 eggs per gram of feces. After 10 minutes, the number of eggs in each chamber was counted under an optical microscope (Labophot-2; Nikon, Japan, objective ×10).

**Table 1. Composition of the different experimental dietary treatments.** (HF: high-fiber diet; HS: high-starch diet; SF: sainfoin pellets supplementation; CONT: control pellets supplementation).

|  | HF-CONT | HF-SF | HS-CONT | HS-SF |
|---|---|---|---|---|
| **Ingredients (% DMI)** |  |  |  |  |
| Hay | 85.8 | 85.8 | 46.1 | 46.1 |
| Sunflower meal | 4.8 | 0.0 | 6.0 | 0.0 |
| Hay pellets | 9.4 | 0.0 | 11.6 | 0.0 |
| Rolled barley | 0.0 | 0.0 | 36.2 | 36.2 |
| Sainfoin pellets | 0.0 | 14.2 | 0.0 | 17.6 |
| **Diets** |  |  |  |  |
| Dry matter intake (DMI) | 2.16 | 2.16 | 1.73 | 1.73 |
| Digestible Energy (DE, kcal/kg DM) | 2169 | 2227 | 2675 | 2748 |
| Crude protein | 8.5 | 8.3 | 11.2 | 11.0 |
| Neutral Detergent Fiber (NDF) | 60.8 | 59.2 | 44.0 | 41.9 |
| Acid Detergent Fiber (ADF) | 38.4 | 38.6 | 26.1 | 26.3 |
| Lignin (ADL) | 5.7 | 6.4 | 4.1 | 5.0 |
| Starch | 0.9 | 0.9 | 21.5 | 21.5 |
| Sugars | 7.0 | 6.8 | 5.8 | 5.5 |
| Crude fat | 1.5 | 1.9 | 1.8 | 2.3 |
| Ash | 5.9 | 5.7 | 4.9 | 4.7 |
| Calcium | 0.6 | 0.6 | 0.5 | 0.4 |
| Phosphorus | 0.2 | 0.2 | 0.3 | 0.3 |

At the end of each experimental period ($D_{21}$), five grams of feces were used for coprocultures. Feces were moistened, mixed with activated charcoal powder, and spread on Whatmann paper (Whatmann 40 CAT No.1440-125) in Petri dishes. These were then stored in the dark at ambient temperature (21˚C) and constant humidity. After 8 days, Petri dishes and Whatmann paper were washed with distilled water and the L3 larvae were collected and centrifuged (1500 rpm for 10 min at 21˚C). The supernatant was discarded, and the motility of the larvae was observed under an optical microscope (Labophot-2; Nikon, Japan, objective ×10). The motility of 50 larvae per horse was assessed according to the following scale: 1 –immobile; 2 –moderately mobile; 3 –highly mobile.

## Analysis of functional bacterial groups

The conventional anaerobic culture technique in roll tubes [32] was used to enumerate total anaerobic, cellulose-utilizing, starch-utilizing and lactate-utilizing bacteria contained in each fecal sample at $D_{21}$. One gram of fresh feces was diluted to decimal in a mineral solution, and inoculated under continuous $CO_2$ flow [33] on a non-selective medium to grow total anaerobic bacteria [34, 35], and on selective media containing starch [36] or lactic acid [37], to grow amylolytic and lactic acid-utilizing bacteria, respectively. After 72 hours of incubation at 38˚C, bacterial concentrations were determined by colony enumeration (Colony Forming Units (CFU) / g of feces). The cellulolytic bacteria were cultured in a complex liquid medium with one filter paper strip as cellulose source for 14 days at 38˚C [38], and concentrations were determined by using the McGrady method of the most probable number (MPN / g of feces). All bacterial concentrations were $\log_{10}$-transformed.

## Bacterial 16S Ribosomal RNA gene sequencing analysis

At $D_{21}$ of each experimental period, an aliquot of fresh feces was sampled sterilely and store frozen for molecular biology analysis. Total DNA was extracted from 0.25 g feces (immediately stored at -80˚C upon collection) following the method described in Yu and Morrison [39]. DNA quantity and purity was assessed using a spectrophotometer (SpectraMax ID3, Molecular devices, San Jose, USA). The V3-V4 hypervariable region of the bacterial 16S DNA was targeted and amplified, sequenced, and analyzed as described by Grimm et al. [40]. Briefly, amplicons of V3-V4 region were obtained after a first round of polymerase chain reaction (PCR) and checked by electrophoresis on a 2% agarose gel. Amplicons were purified and submitted to a second round of PCR and the resulting products were sequenced using an Illumina MiSeq run of 250 base paired ends (Illumina Inc., San Diego, CA, USA) at the Genotoul bioinformatics platform (Toulouse, France). Bioinformatic analyses were performed using FROGS pipelines (Find Rapidly OTU with Galaxy Solution) [41]. In a first step, the obtained sequences were assembled and cleaned. Clustering was then performed using the Swarm method. Chimeras were eliminated and the resulting Amplicon Sequence Variants (ASVs) that were not present in at least four samples or whose abundance was $<5 \times 10^{-5}$ were removed. Remaining ASVs were then aligned to the SILVA 16S database (version 138.13) using BLAST. The abundance table and the associated multi-hit list were generated at the end of this step. The affiliations were then filtered to eliminate ASVs with a percentage of coverage below 99% and a percentage of identity below 90%. Finally, a normalization of ASVs was carried out by reducing all samples to the same number of sequences based on the sample with the lowest number of sequences. One sample collected from one horse in the HF/SF group was removed due to a small number of recovered sequences (2,128 sequences). Richness (number of ASVs and Chao1), diversity (Shannon and inverse Simpson) indices, and Bray-Curtis distance were calculated from the abundance table.

## Assessment of fecal dry matter, fecal pH, and fermentation end-products

One hundred grams of feces were weighed weekly, oven-dried at 70˚C for 72 h and reweighed again to measure fecal dry matter (DM).

Approximately 100 grams of feces were pressed and filtered each week through a 100 μm filter and the filtrates used to measure pH with an electronic pH meter (CyberScan pH 510; Eutech Instrument Europe B.V., Landsmeer, The Netherlands). The filtrates were subsequently frozen (-20˚ C) with a preservative solution composed of 4.25% H3PO4 and 1.0% HgCl2 for later analyses of Volatile Fatty Acids (VFAs), and without preservative solution for lactic acid analysis.

Gas-liquid chromatography (Clarus 500; PerkinElmer, Courtaboeuf, France) was used to analyze the concentrations of total VFAs, acetic (C2), propionic (C3), isobutyric (iC4), butyric (C4), isovaleric (iC5), and valeric (C5) acid, as described by Jouany [42]. We used 4-methyl valeric acid (277827-25G, Sigma-aldrich, USA) as internal standard in all samples, which were injected under nitrogen on a capillary column (Elite-FFAP column; PerkinElmer, Courtaboeuf, France). We computed the concentration of total VFAs as mmol/L of fecal filtrate, and the concentration of each VFA as the proportion of the total VFAs. Additionally, D-lactic acid and L-lactic acid concentrations were determined using a spectrophotometric method at 340 nm with an enzymatic colorimetric kit (Megazyme, D-/L-lactic acid (rapid) Assay Kit, Megazyme International Ireland Ltd., Wicklow, Ireland) following the procedure described by Grimm et al. [43]. Lactic acid concentrations (mmol/L of fecal filtrate) were also expressed as a proportion of the total lactic acid concentration.

## Packed red blood cells, peripheral white blood cells, cytokines, procalcitonin, LPS, and blood acetate

At $D_{21}$ of each experimental period, blood was collected by venipuncture of the jugular vein of each horse.

Hematocrit was measured with a microcentrifuge (Haematokrit 210, Hettich, Germany). For each horse, blood was collected in two capillary tubes and centrifuged at 10,000 g for 5 min. Hematocrit was expressed as the percentage of packed red blood cells (the average between the two measurements).

A drop of blood was smeared on glass slides. Smears were fixed with anhydrous methanol for 1 to 2 min and subsequently stained with Giemsa (10%) (GS-1L, Sigma-aldrich, USA) for 45 min. For each blood smear, 100 white blood cells were enumerated using an immersion microscope (Eclipse E600; Nikon, Japan, under oil immersion × 100), and we computed the percentage of lymphocytes, neutrophils, and eosinophils.

Twenty-four ml of blood per horse were collected in EDTA and dry tubes to obtain plasma and serum. EDTA tubes were centrifuged immediately (1500 g for 20 min at 4° C) and dry tubes were placed at room temperature for 2 hours and centrifuged (1500 g for 20 min at 4° C) to collect serum. Aliquots of plasma and serum were then stored at -20° C for lipopolysaccharide (LPS), procalcitonin, and cytokine analyses.

Plasma of total esterified LPS concentration was determined by direct quantitation of 3-OH C10, C12, C14, C16 and C18 using HPLC coupled with tandem MS (HPLC-MS/MS, Agilent QQQ 6460 device) as described in Pais de Barros et al. [44]. Procalcitonin serum levels were measured using an ELISA Kit for horses (Horse Procalcitonin ELISA kit, MyBioSource San Diego, USA). The preparation of reagents and all incubations and washes were performed according to the manufacturer's instructions. After a 1/10 dilution, plasma acetate concentration was measured spectrophotometrically at 450 nm in a 96-well microplate (AMR 100) using an enzymatic colorimetric kit (Acetate Colorimetric Assay Kit, Merk, Darmstadt, Germany), with the following modification. Immediately after preparation, the microplate was read for the initial absorbance (A0). After adding enzyme to each well, the microplate was agitated on plate shaker at 900 rpm during 40 min to obtain the final absorbance (A1).

To assess the levels of IL-4, IL-6, IL-10, TNF-$\alpha$, and IFN-$\gamma$ in serum, an equine-specific multiplexed cytokine/chemokine magnetic bead kit (EMD Millipore, Merk, Darmstadt, Germany) was used, following the manufacturer's instructions in a 96-well microplate. All samples were run without dilution, and the plates were read using Bio-Plex Multiplex System instrumentation (Bio-Rad Laboratories, Hercules, CA, United States) with Luminex xMAP technology (Luminex Corporation, Austin, TX, United States). A minimum bead count of 50 for each cytokine/chemokine was acquired for data analysis using Bio-Plex Manager Software 6.1 (Bio-Rad Laboratories, Hercules, CA, United States).

## Statistical analyses

We used General Linear Mixed Models (GLMM) to assess the effect of diet and supplementation on FEC and larval motility, functional bacterial groups, microbiota $\alpha$-diversity and richness, fermentation end-products, markers of intestinal integrity and systemic immunity. The models had a normal distribution of errors except for larval motility that was modelled using a multinomial distribution. Some variables were measured several times for each horse within the same experimental period. For these variables, the GLMM included time ($D_0$ indicating the beginning of the dietary treatment for each experimental period), diet (HF vs HS), supplementation (SF vs CONT), and all the two- and three-way interactions as fixed factors. Time was also included as a repeated measure per horse. Horse ID and the interaction between

horse ID and the experimental period were declared as random effects to take into account the non-independence of observations from the same horse.

For variables that were only measured at the end of each experimental period, the GLMM included the same fixed and random effects except for time that was not declared as a repeated measure.

For variables with a normal distribution of errors, GLMM were run using PROC MIXED (SAS). For the larval motility, which follows a multinomial distribution, we used PROC GLIM-MIX (SAS). For all models, the significance threshold was set at $p \leq 0.05$.

We used a principal coordinates analysis (on Bray Curtis distances) to analyze the β-diversity of the intestinal microbiota with the vegan and ggordiplots packages in R. Differences between dietary treatments were assessed with a permutational multivariate analysis of variance (PERMANOVA) with 9999 permutations using adonis function of vegan package in R.

To analyze the abundance of bacterial taxa among the four dietary treatments, we used a linear discriminant analysis effect size (LEfSe) with the Galaxy software package of the Huttenhower Laboratory [45]. This model uses the non-parametric factorial Kruskal-Wallis (KW) sum-rank test to identify taxa with significant differential abundances between groups (using one-against-all comparisons). Finally, linear discriminant analysis (LDA) was performed to estimate the effect size based on a threshold of 3 log LDA scores. Bacterial taxa with different abundances were used to generate cladograms illustrating the effect of dietary treatments.

## Results

### Effect of dietary treatments on strongyle fecal egg count and larval motility

The GLMM provided evidence for a significant three-way interaction between time, diet and supplementation on FEC (Table 2, Fig 2). The number of strongyle eggs excreted by horses in the HF groups did not vary during the 21 days of each experimental period, while the number of excreted eggs increased over time in HS-fed horses. However, the rate of the increase of FEC over time was lower in HS-fed horses supplemented with sainfoin, resulting in a significant three-way interaction.

Each experimental period was separated by a 21-day washout, during which all horses were fed the same hay-based diet. We checked if these washout periods were effective to reset the FEC. If the washout periods were ineffective, we should expect any difference at the end of the

**Table 2. General linear mixed model exploring the effect of diet (HF vs HS) and supplementation (SF vs CONT) on two traits referring to helminth infection (fecal egg count and L3 motility) over the 21 day experimental period.** We report the fixed effects but the model also included the horse ID and the horse ID * experimental period as random effects. FEC were assessed four times during the experimental period therefore the model also included time as a repeated measurement. L3 motility was only measured at day 21 and modelled using a multinomial distribution. We report the degrees of freedom (df), F and $p$ values.

|  | Effect | df | F value | $p$-value |
|---|---|---|---|---|
| FEC | Time | 3,13 | 21.14 | <0.001 |
|  | Diet | 1,30 | 38.90 | <0.001 |
|  | Supplementation | 1,30 | 2.24 | 0.145 |
|  | Time*Diet | 3,13 | 9.80 | <0.001 |
|  | Time*Supplementation | 3,13 | 0.80 | 0.496 |
|  | Diet*Supplementation | 1,30 | 1.79 | 0.192 |
|  | **Time*Diet*Supplementation** | **3,13** | **4.78** | **0.003** |
| L3 motility | Diet | 1,23 | 2.06 | 0.151 |
|  | **Supplementation** | **1,23** | **3.97** | **0.047** |
|  | Diet*Supplementation | 1,23 | 0.13 | 0.715 |

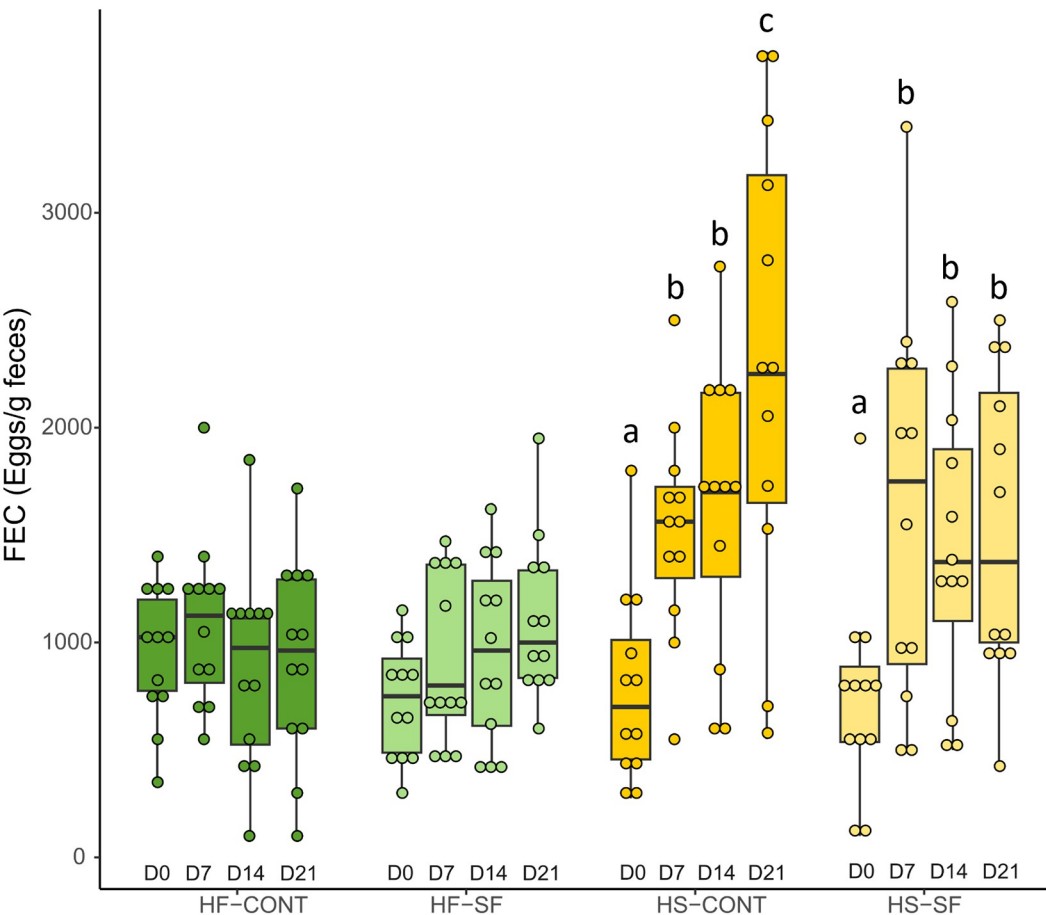

**Fig 2. Changes in FEC over the 21 days of the experimental periods for horses fed high-fiber (green bars) and high-starch (yellow bars) diets and supplemented with sainfoin pellets (light colors) or control pellets (dark colors).** We report the median and the interquartile range. A mixed model ANOVA followed by Tukey's multiple comparisons test show significant differences in the values presenting different superscript letters within each treatment (p < 0.05).

experimental period to be carried over at the beginning of the next experimental period. For instance, FEC increased over time during the first experimental period (signed rank test comparing FEC at $D_0$ and $D_{21}$, $p = 0.0059$). Therefore, in the absence of an effective washout, we should expect FEC values to differ between the $D_0$ of the first and the $D_0$ of the second experimental period. However, a signed rank test showed that this was not the case ($p = 0.1445$). A model comparing the FEC at the four $D_0$ ($D_0$ at period 1, $D_0$ at period 2, $D_0$ at period 3, and $D_0$ at period 4) showed no difference (GLMM: F3,33 = 1.33, $p = 0.2821$) (S2 Fig).

Motility did not differ between larvae retrieved from HF and HS-fed horses (Table 2). However, larvae from sainfoin-fed horses were less mobile than larvae from control-fed horses (Table 2, Fig 3), while the interaction between diet and supplementation was not significant (Table 2).

## Effect of dietary treatments on the diversity and composition of the microbiota

The initial 1,300,442 sequences of 16S rDNA V3-V4 regions obtained from the 47 fecal samples were reduced to 564,555 after quality filtering during bioinformatic processes, with a

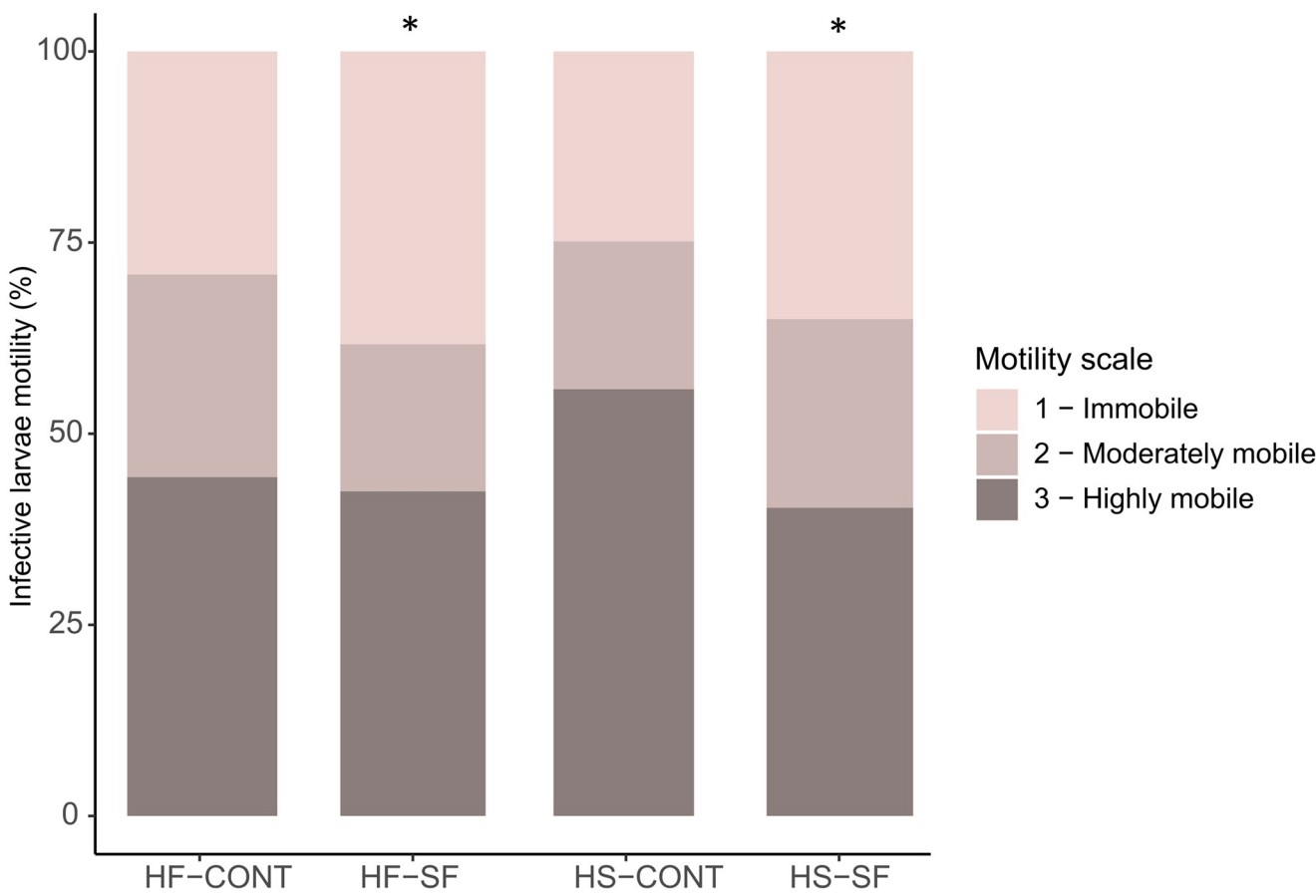

**Fig 3. Percentage of immobile, moderately mobile and highly mobile L3 larvae hatched from eggs recovered at D$_{21}$ from feces of horses fed high-fiber (HF) or high-starch (HS) diets, supplemented with sainfoin pellets (SF) or control pellets (CONT).** Larval motility differed between SF and CONT ($p < 0.05$), independently of the diets.

mean (± SD) of 12,011 (± 3,334) reads per sample. These sequences were clustered into 2,501 ASVs and assigned to 13 phyla, 19 classes, 35 orders, 56 families and 109 genera. The lowest number of sequences retained for the analyses was 3,471 and was used to normalize the data prior to the analysis of microbial richness, α- and **β**-diversity, and relative abundances.

Bacterial richness (number of ASVs and Chao1 index) did not differ between diets nor supplementations (S2 Table, Fig 4A and 4B).

Alpha-diversity as measured by Shannon and inverse Simpson indices was significantly lower for the microbiota of HS-fed horses compared to the HF group (S2 Table, Fig 4C and 4D).

Beta-diversity of the microbiota also differed between horses fed HS and HF as shown by a principal coordinates analysis, and a PERMANOVA model (Fig 5).

Given the high number of unknown species (91%), the analysis of relative abundances considered the genus (37% unknown genera) as the lowest taxonomic level. The LEfSe analysis showed that several bacterial taxa were overrepresented in the microbiota of HF-fed horses compared to HS (Fig 6, S3 Table). At the phylum level, Desulfobacterota and Fibrobacterota were overrepresented in the fecal microbiota of horses fed HF diet, while Proteobacteria were more abundant in the microbiota of HS-fed horses (Fig 6, S3 Table). At the lower taxonomic

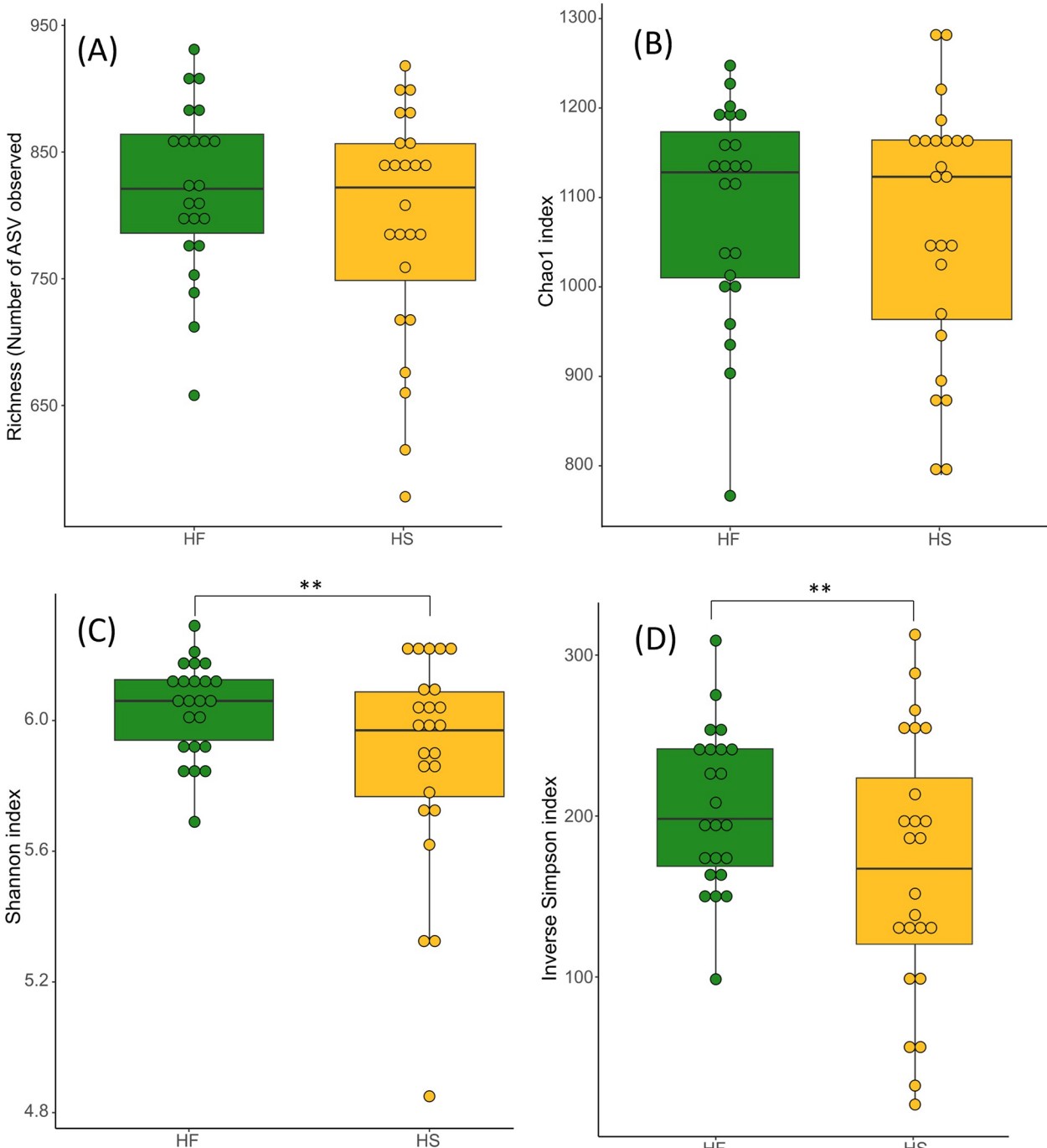

**Fig 4. Fecal bacterial richness (A: number of ASV; B: Chao 1 index) and α-diversity (C: Shannon index; D: Inverse Simpson index) at $D_{21}$ in horses fed high-fiber diet (HF) or high starch diet (HS).** We report the median and the interquartile range. Asterisks indicate significant differences between groups ($p < 0.01$).

level (genus) the relative abundances of 18 genera, including *Fibrobacter*, *Agathobacter*, *Anaerosporobacter*, *Lachnoclostridium*, *Lachnospiraceae AC2044 group*, *Lachnospiraceae NK4B4 group*, *Lachnospiraceae UCG 003*, *Lachnospiraceae UCG 006*, *Lachnospiraceae UCG 008* and *Pseudobutyvibrio* were enriched in the fecal microbiota of HF-fed horses (Fig 6). The fecal

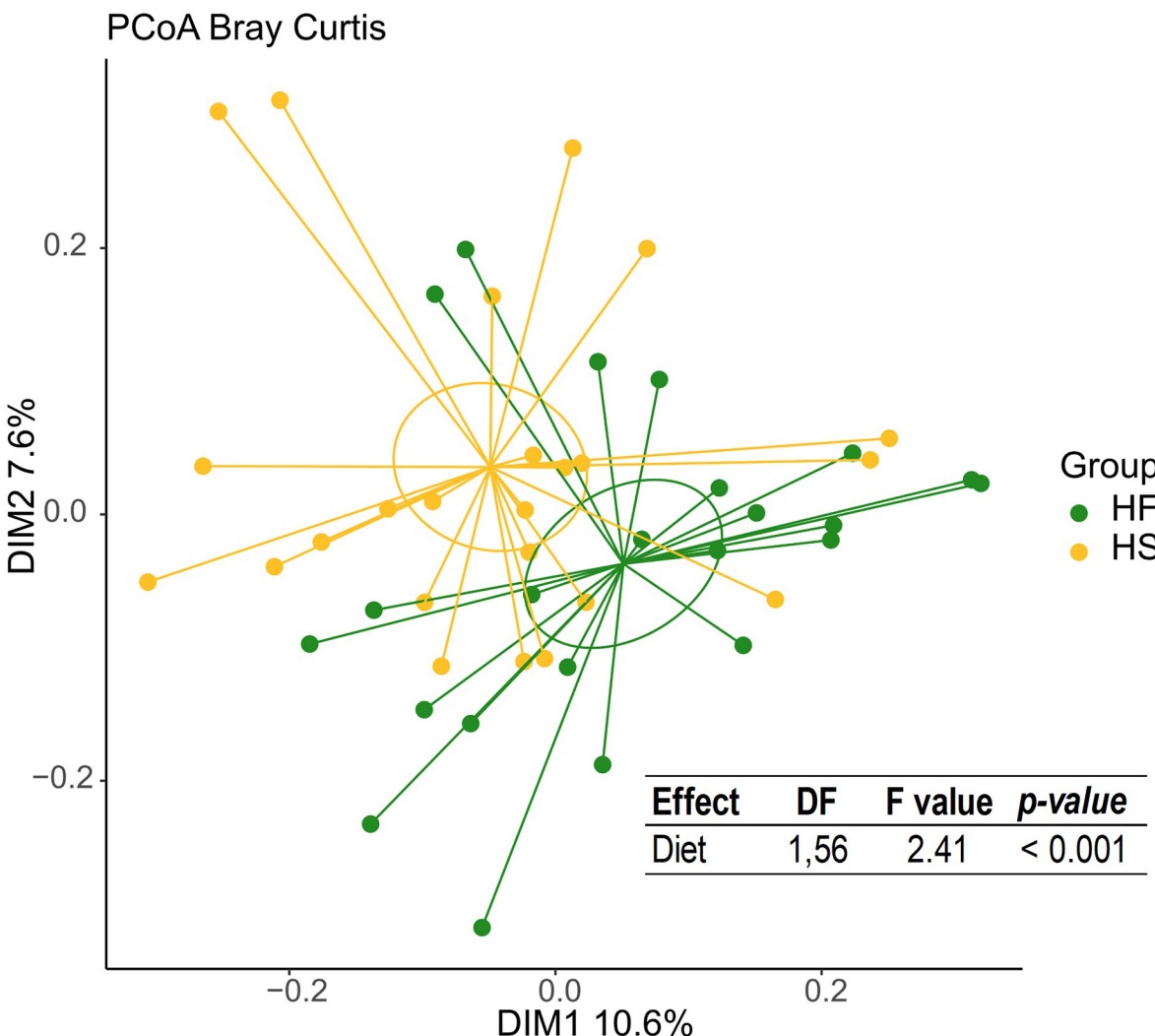

**Fig 5. Principal coordinates analysis (PCoA) at amplicon sequence variants level showing the β-diversity (Bray-Curtis distance) at D$_{21}$ in horses fed fiber diets (HF) or high starch diets (HS).** Ellipses represent the 95% confidence intervals.

microbiota of horses fed HS diet showed an increase in *Monoglobus*, *CAG 352*, *Succinivibrionaceae UCG 002* and a multi-affiliated genus from the *Succinivibrionaceae* family (Fig 6).

The effect of supplementation on bacterial abundances within each diet group was relatively modest (Fig 7A and 7B), with *Eubacterium halii* group and *Noviherbaspirillum* enriched in sainfoin-fed horses within the HF diet (S4 Table, Fig 7A).

## Effect of dietary treatments on microbial functions

The concentration of cellulolytic bacteria (CFU/g of feces) was higher in feces of HF-fed horses, while amylolytic and lactic acid utilizing bacteria were more abundant in feces of HS-fed horses (Fig 8, S5 Table). Neither diet nor supplementation had an effect on the concentration of total anaerobic bacteria (S5 Table).

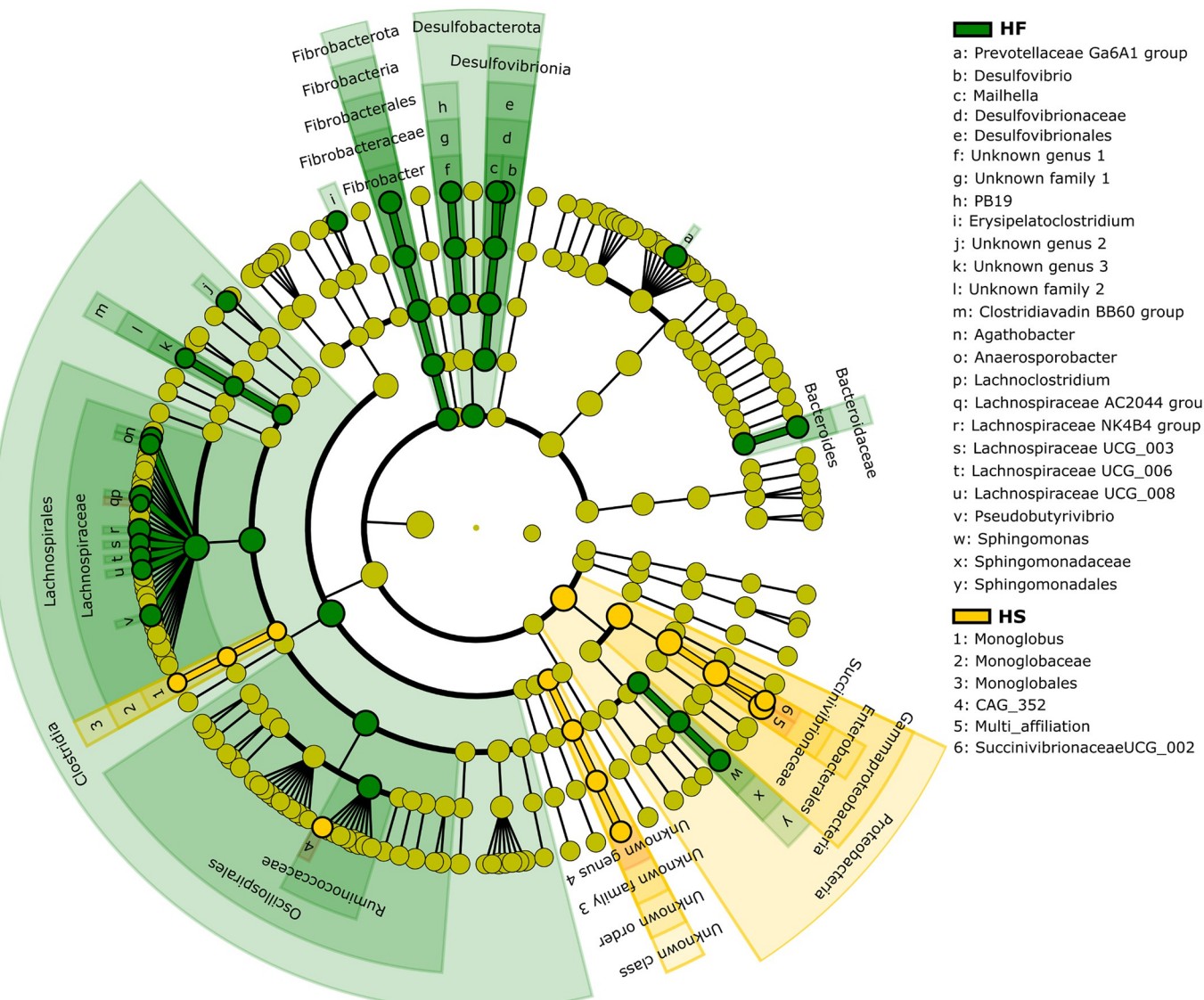

**Fig 6. LDA effect size (LEfSe) cladograms showing the taxa most differentially associated with high-fiber or high starch diet.** The circles represent, going from the inner to outer circle: phyla, class, order, family and genus. Green circles represent taxa overrepresented in HF-fed horses and yellow circles taxa overrepresented in HS-fed animals.

### Effect of dietary treatments on microbial activity

Fecal pH became more acid from $D_0$ to $D_7$ for both HF- and HS-fed horses; however, while pH remained low for HS-fed horses up to $D_{21}$, pH was restored to the initial value in HF-fed individuals (Fig 9). This resulted in a significant time x diet interaction (S6 Table). Supplementation did not affect fecal pH (S6 Table).

Fecal dry matter did not vary, neither as a function of diet nor of supplementation (S6 Table).

Total VFAs were higher in HS-fed horses compared to HF-fed individuals (Fig 10, S5 Table). The proportion of the different VFAs varied depending on the diet, with more acetate

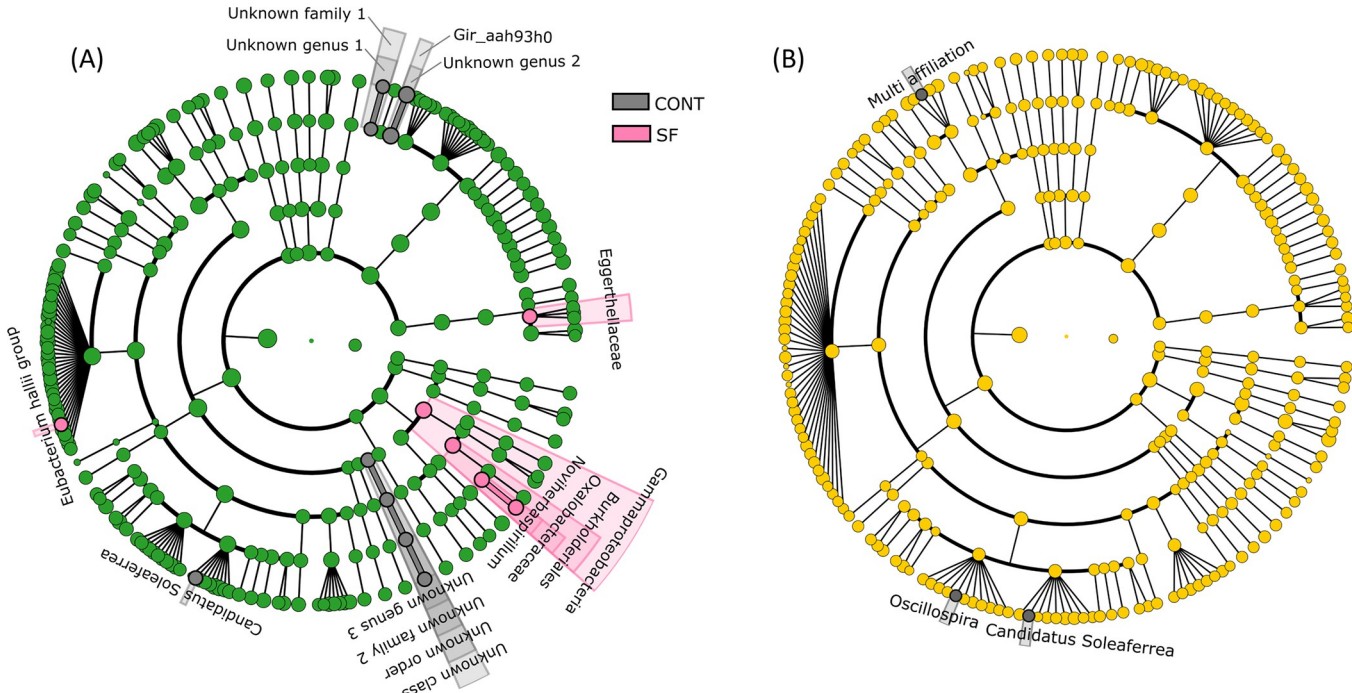

**Fig 7. LDA effect size (LEfSe) cladograms showing the taxa most differentially associated with control or sainfoin supplementation in each diet ((A) High-fiber or (B) High starch).** The circles represent, going from the inner to outer circle: phyla, class, order, family and genus. Pink circles represent taxa overrepresented in SF-fed horses within the HF-diet.

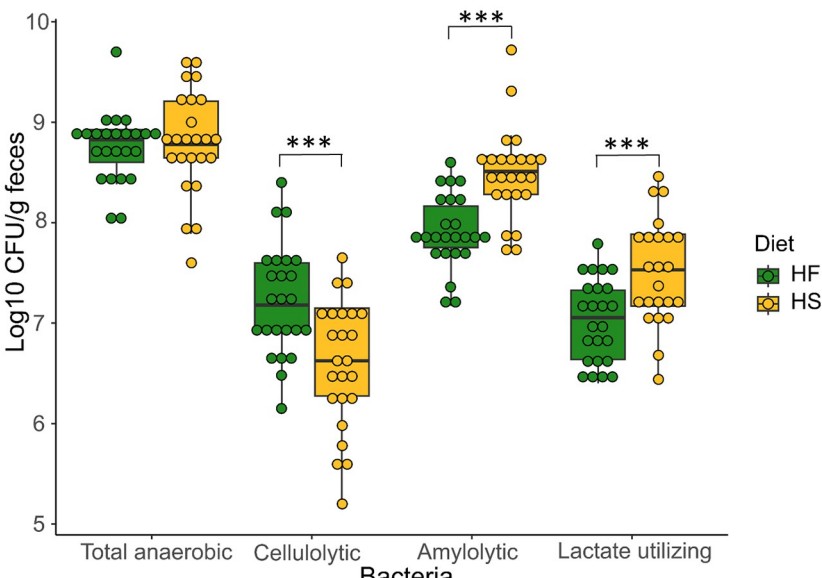

**Fig 8. Total anaerobic, cellulolytic, amylolytic and lactate utilizing bacteria concentrations at $D_{21}$ in fecal samples of horses fed high fiber diets (HF) or high starch diets (HS).** We report median values and interquartile range.

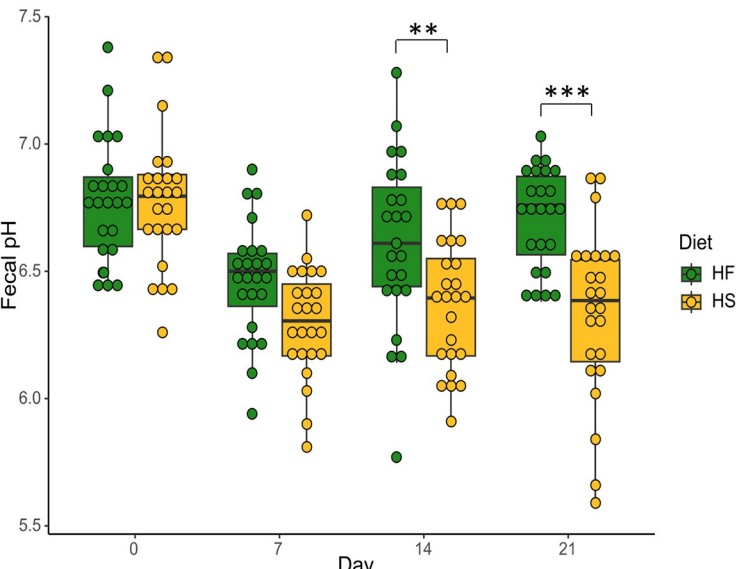

**Fig 9. Changes in pH of fecal samples of horses fed high-fiber diets (HF) or high-starch diets (HS) from $D_0$ to $D_{21}$.**
We report median values and interquartile range. Asterisks indicate significant differences between groups ($p < 0.01$ and $p < 0.001$).

in HF-fed horses and more butyrate, iso-valerate and valerate in HS-fed horses (Fig 11, S5 Table).

The concentration of fecal lactate did not differ neither between diets nor between supplementation groups (Fig 12, S5 Table).

## Effect of dietary treatments on markers of systemic immunity

We used three markers of systemic immunity: the proportion of peripheral white blood cells, the concentration of procalcitonin, and of a panel of Th1 and Th2 cytokines in the serum.

We did not find any difference in the proportion of different white blood cells among dietary treatments (S7 Table). However, including the number of excreted eggs in the model showed a marginally non-significant positive correlation between FEC and the proportion of eosinophils (S7 Table).

Procalcitonin did not differ between diets nor between supplementations (Fig 13A, S7 Table). When including FEC in the model, we did not find any correlation between procalcitonin level and FEC (S7 Table).

We assessed the amount of circulating IL-4, IL-6, IL-10, IFN-γ and TNF-α. For IL-4, only one horse had values higher than the detection threshold (35.88 pg/ml) whatever the diet received. The pattern was relatively similar for the other cytokines, as well. Two horses had values higher than the detection threshold for IL-6, three for IFN-γ, four for TNF-α, and six for IL-10. The same individuals had detectable cytokine levels independently of their dietary treatment and of the number of excreted eggs.

## Effect of dietary treatments on health

We used LPS and acetate circulating in the blood to determine the possible effect of dietary treatments on the integrity of the intestinal mucosa. We did not find any difference in the amount of circulating LPS between diets and supplementation groups (Fig 13B, S7 Table).

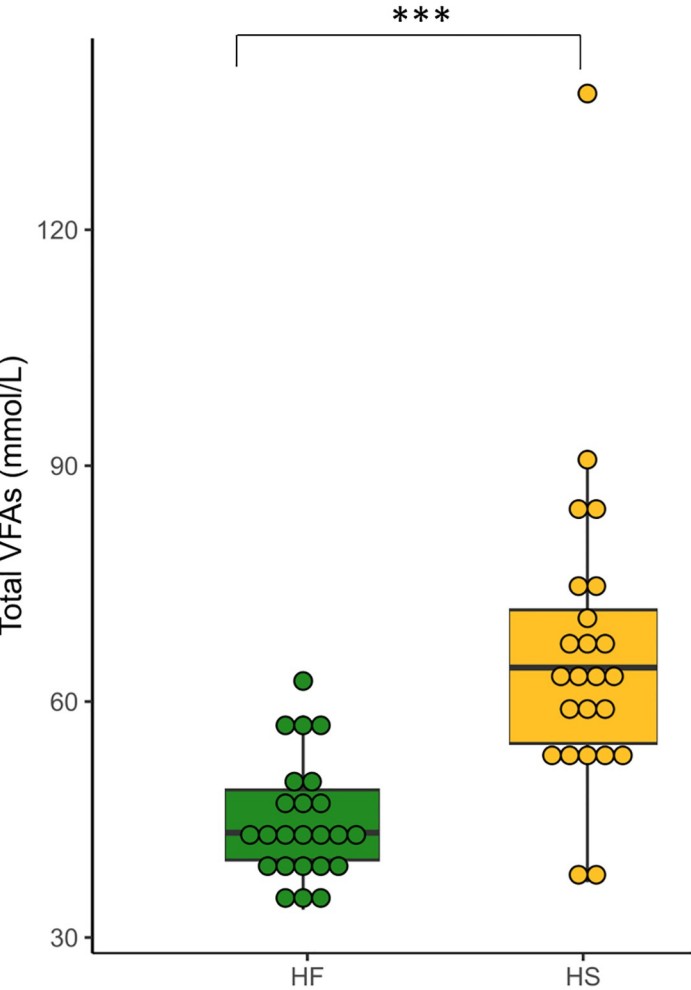

**Fig 10. Total volatile fatty acids (VFAs) concentration at $D_{21}$ in fecal samples of horses fed high fiber diets (HF) or high starch diets (HS).** We report median values and interquartile range.

However, horses in the HF group had higher levels of blood acetate (Fig 13D, S7 Table). Interestingly, the concentration of fecal acetate was higher in HS-fed horses (Fig 13E, S7 Table).

Diet and supplementation had no effect on hematocrit (Fig 13C, S7 Table). We also ran a GLMM with the number of excreted eggs included as a covariate. This model showed that horses that excreted the largest number of eggs also had the lowest hematocrit level (S7 Table, S3 Fig).

## Discussion

We found that horses, naturally infected with gastrointestinal nematodes, fed a high-starch diet excreted more eggs in their feces compared to those fed a high-fiber diet. However, the rate of increase in egg excretion in HS-fed horses was dampened when a polyphenol-rich supplement (sainfoin pellets) was included to the diet. The supplement also affected larval motility, as L3 larvae hatched from eggs recovered in feces of sainfoin-fed horses were less mobile. These results are in agreement with our *a priori* expectation, although the effect of sainfoin on

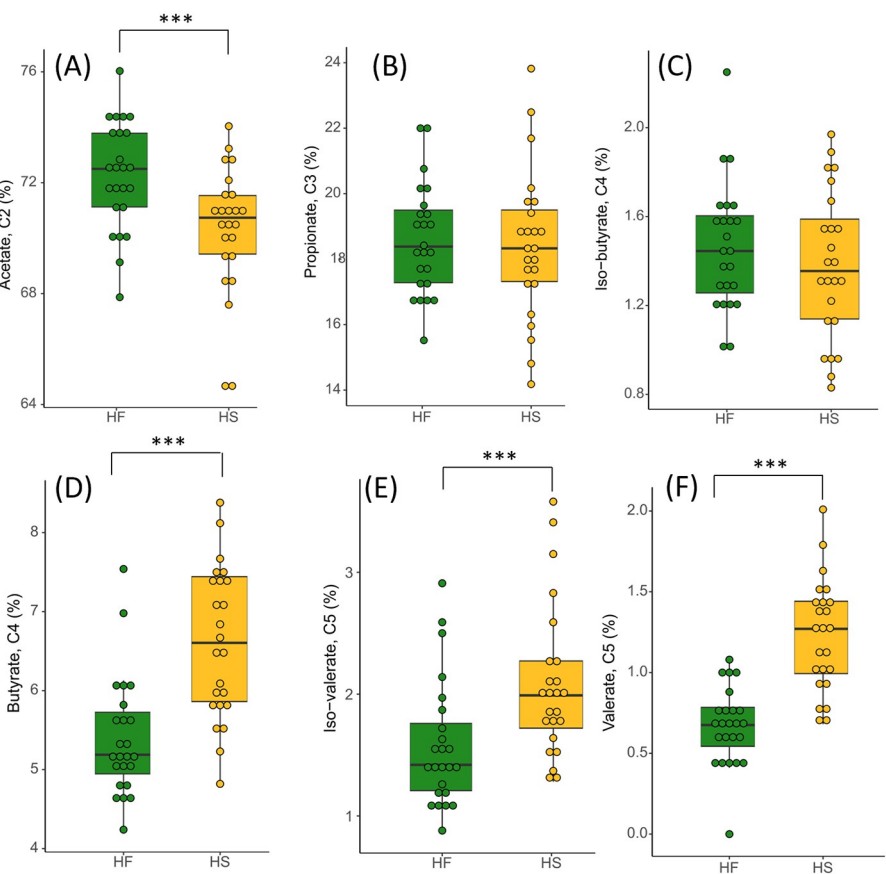

**Fig 11. Proportions of Acetate (A), Propionate (B), Iso-butyrate (C), Butyrate (D), Iso-valerate (E) and Valerate (F) at $D_{21}$ in fecal samples of horses fed high fiber diets (HF) or high starch diets (HS).** We report median values and interquartile range.

egg excretion was restricted to the HS group. Diet also produced extensive changes in both α- and β-diversity of the fecal microbiota, with HF-fed horses having a higher α-diversity and several bacterial taxa overrepresented compared to HS-fed animals. Functions and activity of the microbiota were also affected by the diet: the cellulolytic bacterial functions were reduced in HS-fed horses while the amylolytic bacterial functions were enhanced, resulting in higher concentration of fecal VFAs. These results are also in agreement with the predicted effect of HS diet on the diversity, composition and function of the large intestine microbiota. Finally, contrary to our prediction, we did not find evidence suggesting that diet had an effect on markers of systemic immunity (circulating Th1 and Th2 cytokines, peripheral white blood cells, procalcitonin).

Feeding a HS-diet to horses increased the number of strongyle eggs excreted in the feces over a period of 21 days. An increase in FEC might reflect more adult worms in the intestine or more eggs laid *per capita*. Since our experimental periods only lasted 21 days, we can exclude the hypothesis that the increase in the number of eggs was due to higher infection intensity [the prepatent period being approximately 2 months for cyathostomins [5] and between 6 and 12 months for large strongyles [46]]. In the same line, the finding that FEC was restored to the initial value after a three-week washout period suggests a very dynamical adjustment of egg excretion, more compatible with changes in *per capita* fecundity than in infection intensity. The dietary treatments may have a direct effect on *per capita* fecundity if

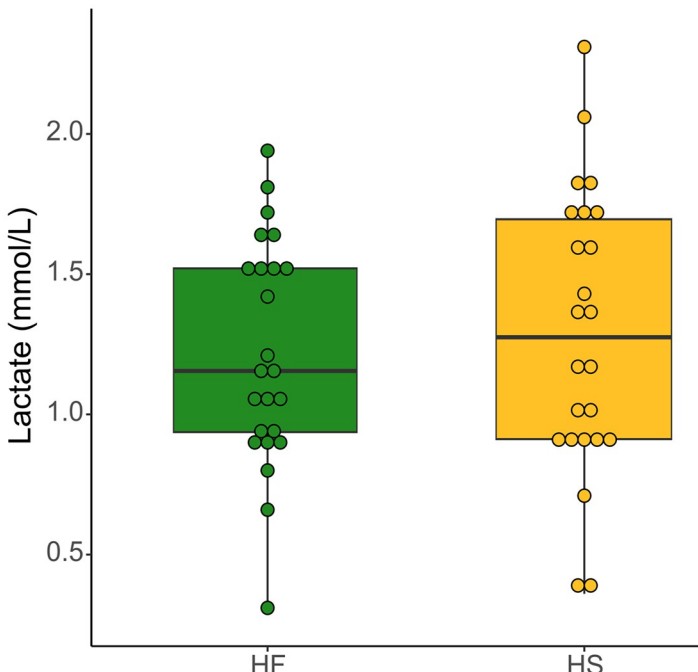

**Fig 12. Lactate concentration at $D_{21}$ in fecal samples of horses fed high fiber diets (HF) or high starch diets (HS).** We report median values and interquartile range.

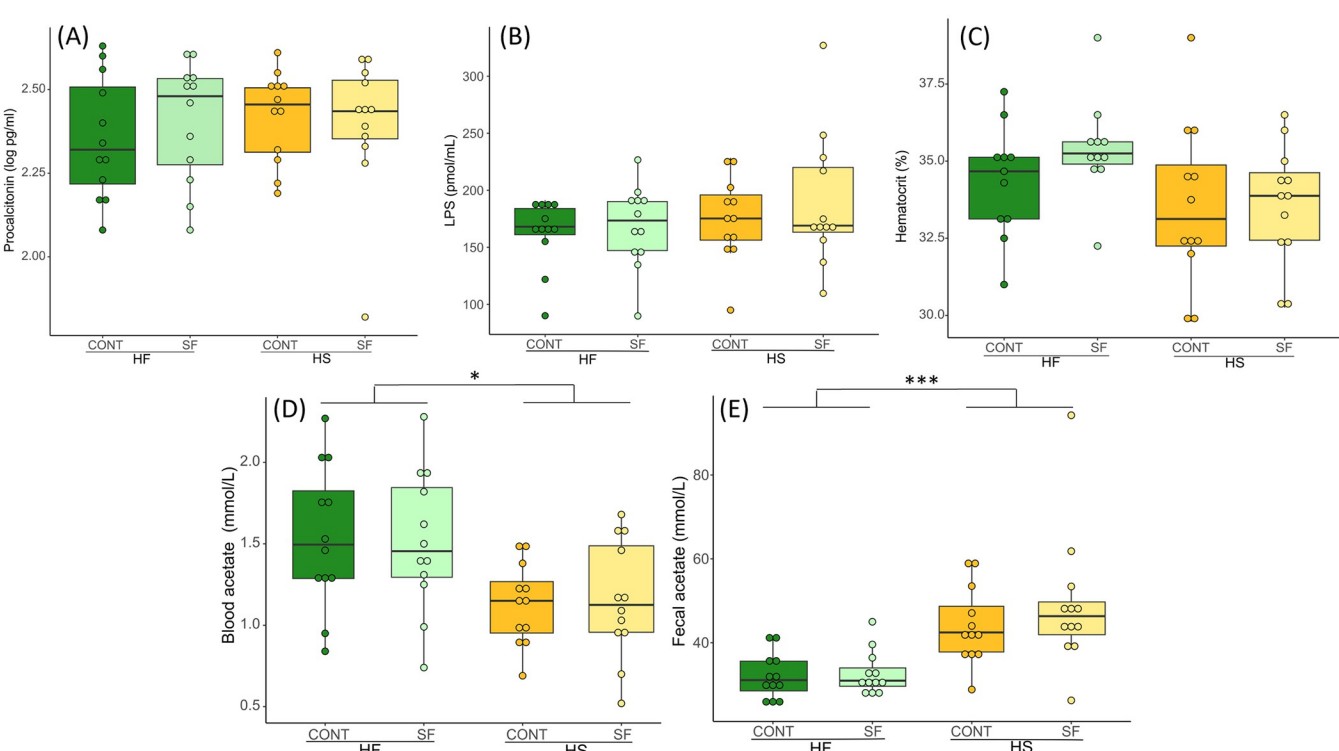

**Fig 13. Procalcitonin concentration in serum (A), LPS concentration in plasma (B), hematocrit (C), acetate concentration in plasma (D) and fecal acetate concentration (E) at $D_{21}$ of horses fed the four treatments (HF-CONT, HF-SF, HS-CONT, HS-SF).** We report median values and interquartile range.

they provide more resources that worms might allocate for egg production. As the experimental diets were iso-caloric and iso-nitrogenous, the increase in fecundity of worms in HS-fed horses could not be due to higher calories or nitrogenous uptake [47, 48]. Strongyles are hematophagous and increased blood glucose in HS-fed horses [49] could provide more readily accessible resources for nematode egg production. Here, we found a negative correlation between FEC and the volume of packed red blood cells that is consistent with the idea of a possible link between the blood meal taken by the worms and their *per capita* fecundity. However, other studies have reported different results regarding the link between diets (carbohydrate-based vs fiber-based diets) and helminth *per capita* fecundity, as for instance in pigs infected with *Oesophagostonum dendatum* [50] and in chickens infected with *Heterakis gallinarum* [51].

In addition to a possible direct, resource-based, effect of starch on parasite fecundity, starch can also change the intestinal ecosystem and the enhanced *per capita* fecundity might be due to these environmental changes. We found that a high starch diet induced a loss of α-diversity with several bacterial taxa being less abundant in HS-fed horses. Similarly, the proportions of the different VFAs produced by the microbiota were largely affected by the dietary treatments. Microbial fermentation resulting from starch ingestion, associated with a reduction of pH, might therefore represent a key environmental trait underlying the increase in the number of excreted eggs. A similar result was reported in chickens infected with the nematode *Heterakis gallinorum*, where diets producing a reduction in the caecal pH are associated with an increase in *per capita* fecundity [51]. However, in pigs infected with *Oesophagostomum dentatum*, a feeding regime based on carbohydrates has been reported to decrease parasite fecundity [52]. In a follow up study on the same model system, intracaecal infusion of Short Chain Fatty Acids and lactate consistently produced a reduction of female *per capita* fecundity [53]. Despite the heterogeneity that exists between study systems, current evidence suggests a possible link between the gut microbiota and helminth infection (establishment, survival, fecundity) mediated by the substrate fermentation and leading to a modification of the abiotic parameters (e.g., pH) of the gut ecosystem [54]. Further studies are, however, needed to better understand the mechanisms involved.

Immunity against helminths is mediated by a type-2 immune response, involving the production of cytokines such as IL-4 and IL-13 which induce the recruitment and activation of immune cells such as eosinophils and mast cells at the intestinal level [55, 56]. We assessed a few systemic markers of immunity and compared them among the dietary treatments. The analysis of type-1 (IL-6, TNF-α, IFN-γ), type-2 (IL-4) and anti-inflammatory (IL-10) cytokines in serum showed that only a handful of samples were above the detection threshold, and the same horses were involved, independently from the diets they were offered. This result suggests that horses included in this study were chronically infected, with relatively low infection intensity [57]. This also shows that the high starch diet did not induce a measurable systemic inflammation, possibly due to the short duration of the experimental periods. Similarly, we did not find any difference in the level of procalcitonin in the serum among diets, consistent with the idea that feeding the horses during three weeks with the high starch diet did not have a pro-inflammatory effect. The proportion of the principal peripheral white blood cells (lymphocytes, neutrophils and eosinophils) was also similar between diets.

A possible health consequence of heavy helminth infection could be a damage of the integrity of the intestinal mucosa leading to an increased permeability and microbial leakage into the blood vessels [58, 59]. Diets producing a more acid environment might represent an aggravating factor, promoting intestinal permeability [43, 60]. We assessed the amount of circulating LPS as a proxy of intestinal permeability [61] and did not find any difference between treatments, showing again that the different dietary groups did not have any direct or indirect

pro-inflammatory effect. However, we also measured the concentration of plasma and fecal acetate. We found that the HS-fed horses have a lower concentration of plasma acetate and a higher concentration of fecal acetate, which suggests that the absorption capacity of the intestinal mucosa was reduced with this diet [62].

The other aim of this study was to evaluate the potential anthelmintic properties of sainfoin, a polyphenol-rich plant. We found that L3 larvae hatched from eggs recovered from feces of SF-fed horses were less motile than larvae from control animals, irrespective of whether they originated from the high-starch or the high-fiber groups. This result is in agreement with previous work conducted both *in vivo* [27] and *in vitro* [63]. The alteration of the L3 cuticle by polyphenols could be an explanation for this loss of motility [15]. The reduction in the motility of infective larvae may affect the infection dynamics, hindering the transmission of the parasite to other individual hosts.

We also found that, within the high-starch diet, the rate of increase of egg excretion with time was lower in horses supplemented with sainfoin, while adding sainfoin did not change FEC in the high-fiber group. These results suggest that any anthelmintic properties of sainfoin on parasite fecundity are context-dependent, namely diet-dependent. In a previous work, a supplementation with sainfoin also resulted in a decrease in FEC, with the reduction occurring mostly after the first week of treatment [27]. Study on ruminants are also in agreement with the possible anthelmintic properties of sainfoin [14, 15]. However, other work has reported no effect of sainfoin on FEC in horses [64, 65]. Given the context-dependence of the effect of sainfoin on FEC, we call for a cautious interpretation of the generality of this anthelmintic property.

We only found minor effects of adding sainfoin on the diversity, composition, function and activity of the microbiota, and on the immune markers considered here. These findings therefore suggest that if any, the anthelmintic effect of sainfoin is not mediated by the microbiota.

## Limitations

This study has some limitations that we would like to discuss here. We only measured systemic immune markers; the next step would be to assess markers of intestinal immunity and how they might change as a function of diet. The experimental periods only lasted 21 days that prevented us from assessing any possible effect of dietary treatments on the establishment and the persistence of the infection. In addition, we only assessed the amount of eggs excreted and the motility of infective larvae but we could not assess the actual intensity of the infection (number of worms). This reflects the difficulties inherent to the model system and the experimental design. Deworming might allow counting adult worms expelled in the feces, but this is not compatible with a Latin square design (where each individual moves from one experimental group to the other). Another limitation associated with the choice of 21-day experimental periods is that any effect of the dietary interventions should be considered as a short-term effect. Since strongyles have long pre-patent periods [5], a long-term study would require follow up over several months. This potentially raise problems related, for instance, to the confounding effect of seasonal variation on the infection dynamics [66].

## Conclusions and future perspectives

Anthelminthic drugs are not a sustainable strategy to control gastrointestinal helminth infections and there is urgent need for alternatives. In grazing animals, the type of diet can have tremendous effects on the intestinal ecosystem, potentially affecting how animals get infected (the establishment of the infection and the excretion of eggs), the microbiota and the intestinal immunity. We found that feeding horses with a high-starch diet rapidly increased the number

of parasite eggs excreted in the feces. This can have a direct effect on the dynamic of the infection, increasing the level of the pasture contamination and improving parasite transmission success. Starch-rich feeds (barley, oat, corn) are commonly included in horse diet, because they are supposed to provide more energy; however, it should be acknowledged that starch-rich food also has a series of undesirable effects on the intestinal ecosystem including helminth infection and microbiota function and activity, as shown in our study. We suggest that the first step in the strategy to control helminth infection should be to provide animals with diets that preserve a healthy intestinal ecosystem. Another important point that we would like to raise here is that pursuing a goal of eradicating helminth infection (zero infection) is not only illusionary but also likely to produce more negative effects than benefits. Mammals have coevolved with helminths during millions of years, and in most cases, the infection does not produce severe symptoms. We therefore suggest that a safer strategy to control helminth infection would be to improve host tolerance to the infection rather than pursuing a hopeless, environmentally toxic, strategy of drug-based eradication. Finally, complex studies in large animals incur substantial costs, which can only be afforded with the commitment of the equine industry and public authorities. This engagement would help bridge the gap between fundamental and applied science to improve animal health and welfare, and preserve the environment.

## Supporting information

**S1 Fig. Experimental design of the study.** Latin square design with four experimental periods of 21 days and 21 days of wash out between periods. (HF: high-fiber diet; HS: high-starch diet; SF: sainfoin pellets supplementation; CONT: control pellets supplementation).
(TIF)

**S2 Fig. Follow-up of individual fecal egg count during the four experimental periods.** The different colors represent the diet allocated to each horse at a specific period. (HF: high-fiber diet; HS: high-starch diet; SF: sainfoin pellets supplementation; CONT: control pellets supplementation).
(TIF)

**S3 Fig. Predicted hematocrit values as a function of fecal egg count at $D_{21}$ across the experimental periods.** Predicted values from a GLMM that that included the effect of the diet, the supplementation and the FEC as fixed effects and horse ID and horse ID * experimental period as random effects.
(TIF)

**S1 Table. FEC (eggs/gram of feces), age (years), body weight (kg) and body condition score (based on the Hennecke scale) of horses included in the study.**
(TIF)

**S2 Table. General linear mixed model exploring the effect of diet (HF vs HS) and supplementation (SF vs cont) on richness (number of ASV's and Chao1 index) and -diversity (Shannon index and Inverse Simpson index) of the fecal microbiota at $D_{21}$.** We report the fixed effects, but the model also included the horse ID and the horse ID * experimental as random effects. We report the degrees of freedom (df), F and $p$ values.
(TIF)

**S3 Table. LDA scores and p values of the discriminant bacterial genera enriched in the fecal microbiota of horses under a high fiber (HF) or a high starch (HS) diet.**
(TIF)

**S4 Table. LDA scores and *p values* of the discriminant bacterial genera enriched in the fecal microbiota of horses under a sainfoin pellets (SF), or a control pellets (CONT) supplementation in each diet (HF and HS).**
(TIF)

**S5 Table. Mixed model exploring the effect of diet (HF vs HS) and supplementation (SF vs cont) on different bacterial functional groups concentrations, Volatil Fatty Acids (VFAs) concentration and proportion and lactate concentartion of fecal samples at $D_{21}$.** We report the fixed effects but the model also included the horse ID and the horse ID * experimental. We report the degrees of freedom, F and *p* values.
(TIF)

**S6 Table. General linear mixed model exploring the effect of diet (HF vs HS) and supplementation (SF vs cont) on fecal pH and fecal dry matter) over the 21-day experimental period.** We report the fixed effects, but the model also included the horse ID and the horse ID * experimental period as random effects. We report the degrees of freedom (df), F and p values.
(TIF)

**S7 Table. General linear mixed model exploring the effect of diet (HF vs HS) and supplementation (SF vs cont) on lymphocytes, neutrophils, eosinophils, procalcitonin, hematocrit, LPS, blood acetate and fecal acetate at $D_{21}$.** We report the fixed effects, but the model also included the horse ID and the horse ID * experimental period as random effects. We report the degrees of freedom (df), F and p values.
(TIF)

## Acknowledgments

The authors are very grateful to all the scientist of Lab To Field who contributed to this work (Cléo Omphalius and Agathe Martin) and to the technical staff of the Plateforme NSP (Manon Guy-Coquille, Charly Poillot and Juliette Choppin de Janvry) for taking care of the horses and helping with sample collection. We are also very grateful to Maria Teixera for her assistance with the lab work.

## Author Contributions

**Conceptualization:** Noémie Laroche, Pauline Grimm, Samy Julliand, Gabriele Sorci.

**Formal analysis:** Noémie Laroche, Gabriele Sorci.

**Funding acquisition:** Samy Julliand.

**Investigation:** Noémie Laroche, Pauline Grimm.

**Methodology:** Noémie Laroche, Pauline Grimm, Gabriele Sorci.

**Project administration:** Pauline Grimm, Samy Julliand.

**Resources:** Samy Julliand.

**Supervision:** Pauline Grimm, Gabriele Sorci.

**Validation:** Pauline Grimm, Gabriele Sorci.

**Visualization:** Noémie Laroche.

**Writing – original draft:** Noémie Laroche.

**Writing – review & editing:** Pauline Grimm, Samy Julliand, Gabriele Sorci.

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
