## [Decision Letter · Decision Letter 0]

22 Feb 2024

PONE-D-23-42676Diet modulates strongyle infection and microbiota in the large intestine of horsesPLOS ONE

Dear Dr. Laroche,

Thank you for submitting your manuscript to PLOS ONE. After careful consideration, we feel that it has merit but does not fully meet PLOS ONE’s publication criteria as it currently stands. Therefore, we invite you to submit a revised version of the manuscript that addresses the points raised during the review process.

We look forward to receiving your revised manuscript.

Kind regards,

Harvie P. Portugaliza, D.V.M., Ph.D.

Academic Editor

PLOS ONE

Journal Requirements:

"The authors declare no conflict of interest. The funders had no role in the design of the study, the collection, analysis or interpretation of the data; in the writing of the manuscript, or in the decision to publish the results."

Reviewers' comments:

Reviewer's Responses to Questions

**Comments to the Author**

1. Is the manuscript technically sound, and do the data support the conclusions?

Reviewer #1: Yes

Reviewer #2: Yes

2. Has the statistical analysis been performed appropriately and rigorously? 

Reviewer #1: Yes

Reviewer #2: Yes

3. Have the authors made all data underlying the findings in their manuscript fully available?

Reviewer #1: Yes

Reviewer #2: Yes

4. Is the manuscript presented in an intelligible fashion and written in standard English?

Reviewer #1: Yes

Reviewer #2: Yes

5. Review Comments to the Author

Reviewer #1: Dear Authors

Concerning your manuscript PONE-D-23-42676 Article Type: Research Article “Diet modulates strongyle infection and microbiota in the large intestine of horses”, I believe it is a very interesting field of applied parasitology and microbiology of domestic and wild equids, namely because horse’s feed/food may promote and/or interfere with both. Besides, since the research on this sense is in need, this paper comes in a good timing, so that countries more prone to horse production can have access to this type of data, since its main conclusions can drive veterinarians, horse producers and owners for a more comprehensive work together, towards a more assertive way of managing horse feeding and its consequences on both gastrointestinal parasitic disease and gut microbiology, which can impact seriously the digestion, gut health and performance.

Besides what will be pointed out, namely that your manuscript has potential to be published, the final decision on the publication of your manuscript at PLOS ONE, depends on the Editor final statement.

Regarding my reviews and comments, they are as follows:

Key-words

As long as there is a specific limit for this item, I suggest adding the word “horse”.

Introduction

Page 3 – You focused too much your attention on the impacts of parasites and their consequences in ruminants. Taken that your manuscript deals with horse parasites, you should write something on this sense, regarding the impact of parasites on their health, production and anthelmintic resistance.

Page 4, Lines 82-84 – I don’t see the need of inserting here this citation regarding a protozoan, a blood parasite, from mice, when previously you mentioned the importance of food in host’s resistance and tolerance to helminths. Therefore I recommend to remove it, because it can also be confusing for the reader.

Materials and Methods

Page 9

Lines 181-182 – It is not clear the way you collect the L3 from the cultures: the Petri dishes were washed? The Whatman filter paper with culture are inverted and placed in a Baermann apparatus? Please be more specific.

Results

Pages 20, Line 428 – Instead of “…Volatil Fatty Acids…”, write “…Volatile Fatty Acids…”.

Discussion

Page 25, Line 551 – Instead of “…infesting larvae…”, you should write “…infective larvae…”.

Limitations

Page 26 – You should give a perspective of an ideal or optimal timespan for a longer field trial like this using a Latin Square Design to assess the same factors in horse parasitism level, both for each period and for the whole extent of the trial, eg., 2 months instead of 21 days and 1 year, instead of 4 months.

It would be interesting also to mention the economical part of such an important set of experiments like these ones you performed with horses, because I feel that available animals and funding is crucial for such a complex study like the one you did. Therefore, perhaps it would be important that you could share your thoughts on this matter, namely if the industry connected with horse production and private horse stud farms should be more engaged in funding this sort of research that is applied, but at the same time fundamental, regarding horse’s health and welfare.

Conclusions

Page 26, Line 590 – Instead of “…level of infection of the pasture…”, write “…level of the pasture contamination…”. The grass/herbage, does not get infected, but just contaminated.

Page 27, Lines 596 – 602 – Although your statements here are very important, this is not a direct conclusion from your work, therefore I advise one of two things: a) Change the title to Conclusions and Future perspectives; or b) Create a a new title after the Conclusions one for Future perspectives and besides other thoughts, move the ones between lines 596 and 602 to this new chapter.

References

Check if all Latin names, whether from parasites or hosts, are well written and/or in italic, because they must. See for instance references 7 and 8.

Best regards and good luck with your amendments.

Reviewer

Reviewer #2: The Manuscript entitled: “Diet modulates strongyle infection and microbiota in the large intestine of horses” is interesting, fits the journal's profile but needs some explanations.

In Introduction is a lack of hypothesis of the study, please add.

I suggest moving the information from l.111 – l.115 to another place in the text, maybe to Material and Methods.

Was the nutritional value of the feed analyzed or are these values calculated (Table 1)?

Please provide detailed information on which microscope the measurements were taken.

What was the internal standard used for the VFAs analyses? What column was used, please provide precise parameters? What was the carrier gas used?

In the discussion, I would suggest more references to current literature.

6. PLOS authors have the option to publish the peer review history of their article (what does this mean?). If published, this will include your full peer review and any attached files.

Reviewer #1: No

Reviewer #2: No

---

## [Author Response · Author response to Decision Letter 0]

8 Mar 2024

Reviewer #1

Key-words

As long as there is a specific limit for this item, I suggest adding the word “horse”.

Response : Done. 

Introduction

Page 3 – You focused too much your attention on the impacts of parasites and their consequences in ruminants. Taken that your manuscript deals with horse parasites, you should write something on this sense, regarding the impact of parasites on their health, production, and anthelmintic resistance.

Response: We agree with this comment. We have added information about the impact of intestinal parasitism on horses' health and anthelmintic resistance in this species. (references added: 4,6, 9,13)

Page 4, Lines 82-84 – I don’t see the need of inserting here this citation regarding a protozoan, a blood parasite, from mice, when previously you mentioned the importance of food in host’s resistance and tolerance to helminths. Therefore, I recommend to remove it, because it can also be confusing for the reader.

Response: We agree with this suggestion, and we removed it.

Materials and Methods

Page 9

Lines 181-182 – It is not clear the way you collect the L3 from the cultures: the Petri dishes were washed? The Whatman filter paper with culture are inverted and placed in a Baermann apparatus? Please be more specific.

Response : We explained this method with more detail in the manuscript (line 181-182 in the revised manuscript with track changes).

Results

Pages 20, Line 428 – Instead of “…Volatil Fatty Acids…”, write “…Volatile Fatty Acids…”.

Response: Corrected. 

Discussion

Page 25, Line 551 – Instead of “…infesting larvae…”, you should write “…infective larvae…”.

Response: Corrected.

Limitations

Page 26 – You should give a perspective of an ideal or optimal timespan for a longer field trial like this using a Latin Square Design to assess the same factors in horse parasitism level, both for each period and for the whole extent of the trial, eg., 2 months instead of 21 days and 1 year, instead of 4 months.

Response: Thank you for this suggestion. We have included a perspective in that direction from lines 587 to 589 in the revised manuscript with track changes.

It would be interesting also to mention the economical part of such an important set of experiments like these ones you performed with horses, because I feel that available animals and funding is crucial for such a complex study like the one you did. Therefore, perhaps it would be important that you could share your thoughts on this matter, namely if the industry connected with horse production and private horse stud farms should be more engaged in funding this sort of research that is applied, but at the same time fundamental, regarding horse’s health and welfare.

Response : Following the referee's suggestion, we have included our thoughts on the importance of the funding of this type of experiments in our “Conclusions and Future perspectives” from line 609 to 612 in the revised manuscript with track changes.

Conclusions

Page 26, Line 590 – Instead of “…level of infection of the pasture…”, write “…level of the pasture contamination…”. The grass/herbage, does not get infected, but just contaminated.

Response: Corrected.

Page 27, Lines 596 – 602 – Although your statements here are very important, this is not a direct conclusion from your work, therefore I advise one of two things: a) Change the title to Conclusions and Future perspectives; or b) Create a new title after the Conclusions one for Future perspectives and besides other thoughts, move the ones between lines 596 and 602 to this new chapter.

Response: The title has been changed to 'Conclusions and Future Perspectives'.

References

Check if all Latin names, whether from parasites or hosts, are well written and/or in italic, because they must. See for instance references 7 and 8.

Response: Done.

Reviewer #2

In Introduction is a lack of hypothesis of the study, please add.

Response: We agree with this comment, and we have included our predictions at the end of the introduction from line 120 to line 123 in revised manuscript with track changes.

I suggest moving the information from l.111 – l.115 to another place in the text, maybe to Material and Methods.

Response: We have merged this paragraph with the predictions at the end of the introduction, which helps understanding the aims of our study.

Was the nutritional value of the feed analyzed or are these values calculated (Table 1)?

Response: The nutritional value of each raw material was analyzed by Dairy One laboratory (Ithaca, USA). We added this information to line 158 in the revised manuscript with track changes.

Please provide detailed information on which microscope the measurements were taken.

Response: We have added the microscope references in the revised manuscript with track changes at lines 177, 184, and 258.

What was the internal standard used for the VFAs analyses? What column was used, please provide precise parameters? What was the carrier gas used?

Response: We have added this information from line 237 to 240 in the revised manuscript with track changes.

In the discussion, I would suggest more references to current literature.

Response: The following references have been added to provide more references to current literature: 48, 54, 56, 57, 58, 59, 60, 61, 62, 65 and 66.

---

## [Editor Report · Decision Letter 1]

25 Mar 2024

Diet modulates strongyle infection and microbiota in the large intestine of horses

PONE-D-23-42676R1

Dear Dr. Laroche,

We’re pleased to inform you that your manuscript has been judged scientifically suitable for publication and will be formally accepted for publication once it meets all outstanding technical requirements.

Kind regards,

Harvie P. Portugaliza, D.V.M., Ph.D.

Academic Editor

PLOS ONE
---

## [Editor Report · Acceptance letter]

29 Mar 2024

PONE-D-23-42676R1 

PLOS ONE

Dear Dr. Laroche, 

I'm pleased to inform you that your manuscript has been deemed suitable for publication in PLOS ONE. Congratulations! Your manuscript is now being handed over to our production team.

Kind regards, 

on behalf of

Dr. Harvie P. Portugaliza 

Academic Editor

PLOS ONE